# Exploring Rotational Grazing and Crossbreeding as Options for Beef Production to Reduce GHG Emissions and Feed-Food Competition through Farm-Level Bio-Economic Modeling

**DOI:** 10.3390/ani13061020

**Published:** 2023-03-10

**Authors:** Alexandre Mertens, Lennart Kokemohr, Emilie Braun, Louise Legein, Claire Mosnier, Giacomo Pirlo, Patrick Veysset, Sylvain Hennart, Michaël Mathot, Didier Stilmant

**Affiliations:** 1Agricultural Systems Unit, Walloon Agricultural Research Centre, 6800 Libramont, Belgium; 2Institute for Food and Resource Economics, University of Bonn, Nußallee 21, 53115 Bonn, Germany; 3INRAE, UMR Herbivores, F-63122 Saint-Genès-Champanelle, France; 4Council for Agricultural Research and Economics, 26900 Lodi, Italy

**Keywords:** beef, climate change mitigation, feed-food competition, innovations, fast rotational grazing, crossbreeding

## Abstract

**Simple Summary:**

Beef production is criticized for its contribution to global warming and its use of human-edible food as feed and hence needs to innovate. Relying on three case studies of beef production systems in Belgium, France, and Germany, we test, using the single-farm model FarmDyn, the interest of fast rotational grazing associated (redesign scenarios) or not (FRG scenarios) to crossbreeding strategies as innovation. The redesign scenarios are adapted to local conditions using early-maturing beef breeds on a French farm or Belgian Blue breeds in a German dairy system and a Belgian suckler cow system becoming, in this last type, a growing and fattening system. Fast rotational grazing induced a higher profit through cheaper feed but an increased workload in pasture management compared to the baseline situation. Beef production from crossbred dairy cows reduces the global warming potential of the systems because of the share of the environmental load with milk production. Crossbreeding with early-maturing breeds, in the French type, has little impact on global warming. The feed-food competition diminished by adapting the stocking rate to the grassland production potential and feeding of by-products. In the future, these simulations should be validated by field trials and a larger diversity of farms.

**Abstract:**

In the context of a growing population, beef production is expected to reduce its consumption of human-edible food and its contribution to global warming. We hypothesize that implementing the innovations of fast rotational grazing and redesigning existing production systems using crossbreeding and sexing may reduce these impacts. In this research, the bio-economic model FarmDyn is used to assess the impact of such innovations on farm profit, workload, global warming potential, and feed-food competition. The innovations are tested in a Belgian system composed of a Belgian Blue breeder and a fattener farm, another system where calves raised in a French suckler cow farm are fattened in a farm in Italy, and third, a German dairy farm that fattens its male calves. The practice of fast rotational grazing with a herd of dairy-to-beef crossbred males is found to have the best potential for greenhouse gas reduction and a reduction of the use of human-edible food when by-products are available. Crossbreeding with early-maturing beef breeds shows a suitable potential to produce grass-based beef with little feed-food competition if the stocking rate considers the grassland yield potential. The results motivate field trials in order to validate the findings.

## 1. Introduction

Cattle, as ruminants, may contribute to food security through the conversion of feedstuff non-edible by humans into high-quality food [1]. However, in recent years, the sustainability of cattle production has been questioned due to adverse effects, such as its contribution to global warming and the enhanced use of feeds potentially eligible for direct human consumption [2]. The production of 1 kg of protein from milk and meat from cattle uses, on average, 0.7 kg of edible protein and results in emissions of 28–640 kg CO_2_eq of greenhouse gases [3]. The reduction of the competition between feed and food production is part of the transition toward more sustainable agriculture, addressing multiple impacts [2,4,5,6].

With 76 million cattle in Europe in 2021, the sector generated a production value of 82 billion EUR. Near half of these cattle is located in France (22%), Germany (15%), Italy (8%), and Belgium (3%) [7]. Given the economic importance of the sector and its contribution to climate change, innovations are needed to adapt production toward a reduced impact on the environment [8].

In this context, the SustainBeef project, gathering teams in Belgium, France, Germany, Ireland, and Italy, aimed to co-define and evaluate sustainable beef farming systems based on resources non-edible by humans. Several innovations to increase the cattle sectors sustainability have been identified using literature review and focus group interviews in Belgium, France and Italy [9]. Two innovations were identified by farmers and advisors as particularly relevant to improve the use of grasslands: fast rotational grazing (FRG) practices [10] and crossbreeding [11,12]. In rotational grazing, pastures are divided into small paddocks, and the herd occupies one paddock for three to five days before being moved to the next paddock in a predefined order [10]. In FRG, the residency time is decreased to 0.5 to 3 days. The benefit of such a fast rotation is a higher grass quantity and quality compared to the more common continuous grazing due to a better composition of forage by less selection [13].

Crossbreeding refers to crossing two breeds benefiting from the heterosis effect [14]. Crossbreeding can produce cattle with higher roughage intake capacity, adapted to raw fodder valorization and higher growth rates, as well as better meat quality. While in Ireland, most of the beef cattle are crossbred [15], in France and Belgium, suckler beef breeds are mainly late-maturing animals with high muscle development needing high amounts of concentrates to reach the right fat grade at slaughter [14].

Several studies have addressed both innovations with different breeds and regional settings (e.g., [5,16]). A combined analysis and the possible inclusion in existing production systems is, to our knowledge, under-researched.

Mathematical programming models are a common approach to investigating the impact of innovations at the farm level and assessing the sustainability of agricultural production (e.g., [17]). Models at the farm scale, such as the FarmDyn model, focus on analysis at the farm level and are frequently used for assessing environmental and economic impacts [18]. The focus on the farm level as the key decision-making unit allows for capturing the impacts of management scenarios and farmers’ adaptation to changing conditions [19,20]. The advantages of optimization models can also be used in large-scale sensitivity analysis, which is important in light of high biological variability in key factor assumptions [21,22].

In this study, we use FarmDyn to assess the impact of the introduction of FRG and crossbreeding in typical European beef production systems on their contribution to climate change, protein production efficiency, work time, and farm profit [3]. To account for the high biological variability, a sensitivity analysis is performed to test the impact of key assumptions on the results. The goal is to inform farmers and policymakers about the bottlenecks and benefits of the inclusion of FRG and crossbreeding in order to increase the sustainability of current production systems.

## 2. Material and Method

The analysis is performed from cradle to farm gate based on data from one year (2017), covering representative farms from Belgium, France, Italy, and Germany. The identification and gathering of farm data was part of the SustainBeef project [6]. The farms are also used in [6,23] and are considered to be typical farms in major production regions.

### 2.1. The Three Beef Production Systems

Three systems are tested in two scenarios compared to a baseline. Key characteristics of the three baseline systems and the included farms in each system are summarized in Table 1.
-The first system articulates two Belgian (BE) farms. The first one is an integrated crop-livestock farm. It holds a suckler cow herd of the Belgian Blue breed and sells the weaned male offspring, cereals, and sugar beet. The weanlings are transferred to a second farm in the system where they are fattened indoors using maize silage and concentrates as feed;-The second system starts on a suckler cow farm located in the Massif Central, France (FR-IT-B). The farm keeps a herd of Charolais and Salers cows. The herd valorizes pastures during the summer and is kept indoors during the winter. The weaned calves are shipped to a second farm in Italy, where they are fattened indoors using maize silage and concentrates for feeding;-The third farm is an integrated crop-dairy farm fattening its own male offspring of the Holstein breed. The animals are fattened indoors using maize silage and concentrates as feed. Besides cattle production, the farm is also involved in cash-crop production (cereals and sugar beet).

### 2.2. Scenarios

An overview of the scenario design and affected farms is given in Figure 1. Three systems consisting of the five farms are tested in two scenarios and compared to a baseline.

In the first scenario (FRG), the farms having grasslands can manage these with fast rotational grazing. Fast rotational grazing refers to cattle periodically being moved among paddocks as opposed to continuous grazing, where a single plot is grazed for the entire season. Previous studies have found that FRG can improve the quantity and the quality of the grazed grass [24]. The increase in yield is due to a more evenly grazed sward, and an optimal balance of regrowth and grazing to offer fresh grass with high nutritive value. FRG therefore has consistently higher forage yield with a comparable yield distribution [25]. The choice of applying FRG, the area of grassland impacted and the cattle type to which it is applied results from the economic optimization performed with the model and are not defined a priori.

The affected farms are the breeding farms in BE and FR-IT and the dairy herd on the farm in GE. Due to the lack of data on actual FRG on the case study farms, possible yield distributions have been derived based on the yield distribution from grazing observed in the baseline with the help of field experts. The FRG has also been discussed and validated in focus group interviews with farmers [9]. The approach was chosen to ensure that yields reflect conditions faced by farmers and not laboratory conditions.

The resulting monthly dry matter (DM), crude protein (CP), and metabolizable energy (ME) yield for continuously grazed and FRG grazed pastures in each system is depicted in Figure 2. The yearly total DM yield of FRG is 9.0 t/ha in BE, 4.4 t/ha in FR, and 9.9 t/ha in GE compared to 8.0, 4.0, and 9.0 t/ha, respectively, in the baseline. The total crude protein yield per year of FRG is 1.9, 0.9, and 2.1 t/ha in BE, FR-IT, and GE, respectively. The grazing period and yield distribution is determined by local pedo-climatic conditions with FR-IT having the lowest yield due to the mountainous climate and poor soils. Supplementary feed on pastures is optional for the model to meet the animals’ nutrient requirements. The practice is bound to higher work time requirements and costs due to extra fencing and herding. The work time for pasturing increases by 10%, and the variable costs increase by EUR 37.5/ha compared to continuous grazing.

The second scenario redesigns the systems (SR) to promote the transition toward more sustainability. The scenarios are designed by combining crossbreeding and FRG considering local conditions. Field experts and focus group interviews with advisors and farmers initialized the scenarios. The possibility to use FRG is carried over from scenario 1.
-In the BE system, the fattening farm is removed from the system while the breeding farm in the system is transformed into a growing-fattening farm. A dairy farm is added to the system to supply male calves entering the growing-fattening farm. The dairy farm keeps a herd of 70 cows of the Holstein breed. For the renewal, some cows are inseminated with female Holstein-sexed semen while others are inseminated with male-sexed semen of the Belgian Blue breed. The calves enter the growing-fattening farm when 3 weeks old at the cost of EUR 200. They are raised and fattened on grasslands and repurposed stables of the initial suckler cow enterprise. The bulls are grazing from April to October. Based on the performances observed by [11,26], the bulls are sold when 19 months old with a carcass weight of 330 kg (carcass yield 55%), a conformation score of U-R, and a fat score of 2–3. A price of EUR 3.4/kg carcass was therefore considered based on the official Belgian prices [27];-In the redesigned scenario of the FR system, the fattening farm in Italy is removed from the system, and the finishing period is happening at the French breeding farm. Furthermore, the Charolais breed cows are inseminated with the Angus breed. The Angus breed is known for its ability to valorize grass resources and the high fatness score of its meat [28], resulting in higher selling prices. The bulls are slaughtered when 14-month-old at a carcass weight of 300 kg with a price premium of EUR 0.4 per kg carcass weight, adding up to a total price of EUR 4.18/kg carcass;-In the German redesigned scenario, the farm uses crossbreeding and sexed semen to reduce the number of breeding heifers to the herd renewal needs and produce high-yielding Belgian Blue by Holstein crossbred male calves. The calves are fattened at the farm and sold at the age of 21 months at a carcass weight of 413 kg and a price of EUR 3.8/kg carcass.

### 2.3. Overview of the FarmDyn Model

The FarmDyn model has been used for sustainability assessments in the context of European cattle farming before (e.g., [22,23,29]). Here the model is used to analyze each farm in each scenario, over the period of one year. FarmDyn is a bio-economic single-farm optimization model [30]. The model maximizes farm profits by optimizing agricultural activities subject to boundary conditions given by farm endowments, prices, legal restrictions and policies, and available technology.

Farm activities include farmers’ decisions: the type and quantity of livestock to keep, their feeding, the crops cultivated and the fertilization associated. Herd demographics including calf birth, raising, replacement, slaughter, and selling are captured monthly. The methodology of the feed planning tool Zifo2 [31] is used to estimate the animals feed requirements, by considering animal performance and lactation periods balancing the needs for dry matter, fiber, protein, energy, and nutrient intake. Feeding can be performed by a variety of bought and self-produced feedstuff described in [32].

The cropping activities are scenario and farm-specific with regional yields and land endowments. Arable crops are divided into cash crops (wheat, barley, rapeseed, sugar beet) and fodder crops (maize silage and catch crops).

Grassland production options consider different forms of harvest (silage and hay in bales or pit and grazing) and different cuts differentiated by seasonality, productivity, and quality of the harvest. The fertilizer needs of crops and grassland are calculated by the model by considering the total nutrient removal in the form of harvested products, nutrients delivered from soil and air, and leached and gaseous nutrient losses. The fertilizer need can be met by manure, excreta from grazing animals, and mineral fertilizer.

Manure is handled as solid (BE, FR-IT) or as liquid manure (GE). Manure is reused for crops and grassland fertilization within the farm. In dairy herds, milk is considered, for the analysis, as a by-product. Economic allocation is applied to allocate the impacts between the production of milk and beef. It is the preferred method for allocation because the necessary information on prices and economic flows is readily available to be used in the modeling framework. When available, the prices are taken from the farm data. Where no exogenous market price exists, the optimization model is used to provide the shadow price (The shadow price refers to the opportunity costs and are derived as a marginal value in the optimization process) for the economic allocation [33].

### 2.4. Sustainability Indicators

The systems’ contribution to greenhouse gas (GHG) emissions, the work time (WT) invested in production, farm profit, and the human-edible protein production efficiency (HEP) are calculated as sustainability indicators. A Life Cycle Assessment approach similar to [23] is used for GHG emissions and WT.

The system boundaries, as depicted in Figure 3, to calculate these indicators include all the processes, from cradle to the farm gate, necessary to deliver 1 kg of beef carcass, considering culled cows, heifers, and bulls. This constitutes the functional unit in which the WT and GHG emissions are expressed in each system.

Included processes are crop production for feeding with the substages cultivation, seeding, fertilizing, pesticide application, liming, and harvest, herd management including caretaking of cows, heifers, and calves, fattening, and transport of animals from one farm to another in the French–Italian and the Belgian systems.

Agricultural inputs and services are included within the system boundaries, i.e., machinery production and operation, energy, concentrate, fertilizer, and pesticide productions.

Estimated GHG emissions include methane (CH_4_), nitrous oxide (N_2_O), and carbon dioxide (CO_2_). The methodology used and considered sources to quantify on-farm emissions are from Table A1 in the Appendix A. Emissions arising from the provision and transport of major externally bought inputs and services, namely feedstuff, bedding material, fertilizers, pesticides, diesel and agricultural machinery provision and operation, are taken from the Ecoinvent database version 3.6 [34]. GHG emissions are characterized using the global warming potential (GWP) in kg CO_2_eq. with the ReCiPe methodology at the midpoint level (hierarchist perspective) [35].

The WT indicator considers the time taken for herd feeding and caretaking, calving, fieldwork allocated to beef production, stable maintenance, fertilization, management, and administrative work.

Farm profit is calculated considering the following costs and revenue streams: revenues from sold beef (old cows, heifers, bulls), sold milk, sold calves and cash crops, subsidies (coupled and decoupled support), and costs from animal replacement, calf rearing, costs of bought feed, costs of fertilizers, phytosanitary products, diesel purchases, variable machine costs (maintenance and operation), other variable costs (veterinary interventions, crop insurance, etc.) and depreciation of basic structures (sheds, silos) and machinery (tractors and applications). Data on prices and work time requirements of different tasks have been collected in the respective farms. Missing data on prices and work time was complemented from [36]. The profit is computed at the farm level, in the farm implementing the innovate practice: the breeder farm in BE and FR-IT and the dairy farm in GE.

The net HEP efficiency is calculated as the share of human-consumable protein produced in the form of beef meat divided by the amount of human-consumable protein in fodder fed used to produce beef meat. The human-consumable share of protein and calorie content of the feedstuff and meat is based on Laisse et al. [5], Ertl et al. [4], and Wilkinson [37]. The indicators represent the contribution of beef production to human nutrition [6].

In the beef production systems, where breeding and fattening happen in different farms, impacts of beef meat produced in the breeding farms are calculated per kg of transferred animals, which are subsequently implemented as emission factors into the optimization problem of the fattening farm in order to assess the whole system’s performance.

### 2.5. Sensitivity Analysis

The sensitivity of results to key assumptions on parameter values is tested in a global all-at-once sensitivity analysis. This includes the capacity of sheds to contain animals, the yield of grasslands utilized for FRG, and the age at which the bulls are slaughtered. Parameter values vary by ± 20% for the stable size and by 10% for grasslands yields in FRG and for the age at which the bulls are slaughtered from the median scenario. Using Latin Hypercube Sampling, a sample of 100 draws with simultaneously changed levels of the parameters is created, covering the full range of possible factor level permutations. Uniform distributions without correlations are assumed. For each draw and each farm, the profit optimization is run again, considering the changed parameters. The results of each optimized farm are combined in a single data frame for each system. GWP, net HEP efficiency, WT, and the farm of interest profit in each scenario are compared to the baseline. The net HEP efficiency is then studied as a function of the stocking rate, defined as the livestock unit (LU) per ha of permanent grasslands, as the other surfaces resources could potentially produce human-edible food.

## 3. Results

### 3.1. FRG Scenarios

The effect of FRG on GWP is marginal (< ±2%), whatever the system (Figure 4 and Figure 5). The better nutritive value of the FRG swards reduces methane emissions from enteric fermentation (−6% in BE). However, emissions related to grassland and crop management (+10% in BE) are higher because of the higher fertilization, including through animal dejections, required to support the higher quality and quantity yield of FRG.

In the BE case, the GWP improves by 1% (−0.3 kg CO_2_eq/kg beef) compared to the baseline. The additional yield from FRG reduces the area of maize needed for feed production. The profit is increased (+5%) thanks to the freed-up land that is used for cash-crop production (wheat, sugar beet). The impact of innovation on farm input/output is summarized in Table A2 in the Appendix A. The introduction of FRG increases working time to produce 1 kg of carcass by 17.5% (1.10 min per kg carcass) compared to the baseline. The net HEP efficiency in BE in the FRG scenario is 0.69. This is a slight increase of 3% compared to the baseline related to the substitution of maize silage and concentrates, including protein that could be used for human nutrition, by grazed grass. For both the baseline and the BE-FRG, the sensitivity analysis shows a net efficiency negatively correlated with the stocking rate in the breeder farm and increased further under 3 LU/ha of permanent grassland (Figure 6).

In the FR-IT case, the higher emissions from fertilizer application outweigh savings in enteric fermentation resulting in a slightly higher GWP compared to the baseline (+0.6%, +0.2 kg CO_2_eq/kg beef). The introduction of FRG management improves farm profitability in comparison to the reference scenario (+4%). The invested time to produce a 1 kg carcass increased by 18.9% (1.22 min). The net HEP efficiency improves by 4.5%, which results in an indicator value of 0.6. The sensitivity analysis reveals that the net HEP is limited at 0.6 for a stocking rate under 1.1 and drops for a higher stocking rate (Figure 6).

In the GE case, the FRG is mainly used for dairy stock. The innovation has, therefore, a limited effect on beef production. The farm profit increased by 2.5%. The GWP is reduced by 1.6% (−0.2 kg CO_2_eq/kg beef) compared to the baseline. Besides the effects observed in the BE and FR-IT cases, the additional yield in the GE case is used to replace bought concentrates that bare a high emission load from production. The work time to produce 1 kg of carcass is increased by 9.5% (0.3 min), and the net HEP efficiency is slightly reduced by (−0.4%). The sensitivity analysis shows no significant dependence on the stocking rate (Figure 6).

### 3.2. SR Scenarios

In the BE-SR scenarios, the farm of interest profit is improved by EUR 48,000 (+28%). While the production cost is increased by EUR 67k, which mainly includes the costs of the crossbred veal and additional feed (pressed sugar beet pulp) costs, the subsidies are reduced by EUR 17k, corresponding to the coupled subsidies associated with the suckling cows of BE-base. The revenues are increased by EUR 84k for beef and EUR 45k for crops. The quantity of beef produced is about 83,691 kg carcass for a stable of 300 bulls per year, which is twice the baseline situation and contains only finished meat. At the system level, the GHG emission per kg of carcass produced is reduced to 13.2 kg CO_2_eq/kg carcass (−52%). The net HEP efficiency is increased (Figure 5) and negatively correlated to the stocking rate according to the sensitivity analysis (Figure 6). Values above one are obtained thanks to the stable size considered, the use of grass, and bought by-products for the bulls fattening.

In the FR-IT-SR system, the redesign requires keeping animals longer on the breeding farm, evolving toward a breeding and fattening farm. As each animal requires more feed, fewer cows and bulls are produced compared to the baseline system, but more beef carcass is produced (19,317 kg carcass (+17%) of finished meat, in the median run), leading to higher beef revenues (+14%) and profit (+20%). Considering the whole production cycle with the initial Italian fattening farm, this redesigned system maintains (−0.5%) the global warming potential of beef meat. The sensitivity analysis shows that its net HEP efficiency is negatively correlated with the stocking rate, and a stocking rate lower than 1.1 enable a protein net efficiency higher than 1. This system is less labor efficient relative to the baseline situation but more labor efficient than in the FRG scenario.

In the GE-SR case, the crossed bulls tend to be more profitable (+1.5%) and have nearly no influence on climate change (−1%) as they have a higher live weight gain and carcass yield. Due to the use of sexed semen, the number of bulls fattened is higher, which globally increases beef production to 40,629 kg carcass (+13%). This leads, compared to the initial scenario, to an increase in the demand for protein feed. The additional animals fattened and the higher protein demand for fattening lead to an extension of fodder production on arable land, i.e., maize silage, and a higher import of protein-rich concentrates. Therefore, the net HEP efficiency declines to 0.4 (−20%), as more forage comes from feed that is in competition with humans with no stocking rate dependence, as for the FRG scenario.

## 4. Discussion

### 4.1. Fast Rotational Grazing

Defining baseline grazing practices in Belgium, Germany, and France in beef production is difficult due to the diversity of the grazing management schemes implemented and the lack of available literature relating to the impact of grazing management on forage yields, both in quantity and quality, in commercial farms. The qualities considered in our study are lower than those found in [38] in July–October and are, therefore, potentially pessimistic compared to current farmer practices.

Still, at the farm level, the main impact of FRG is an increase in work time due to the increase in the workload related to pasture management. Since pasture complementation is possible and performed, no significant time reduction is observed for feeding. Nevertheless, this supplementation depends on farmer objectives, and some testimonials state that working comfort is improved thanks to reduced time spent indoors for animal management [22,25]. The second effect is the reduction of tillable land used for feed production, resulting in higher cash-crop production and, therefore, profit. While expected, this observation deserves further investigation in regions where droughts have become more frequent in the last years, leading to the use of more conserved fodder. Indeed, only a global grass yield uncertainty has been included in the sensitivity analysis, but no strong grass growth limitation is sometimes observed under extreme drought periods.

Applying the FRG to breeder farms only shows globally a limited impact at the system level. Indeed, the net HEP efficiency evolution is limited by the use of feed in competition with humans during the fattening process. Concerning enteric methane emission in this scenario, while the effect of an increase in grass quality on cattle ingestion is taken into account, the potential reduction in methane production rate is not considered in this study since we found no coherent Tier-3 methodology that could account for this effect. This effect deserves further exploration.

Since fertilization is computed based on nutrient removal, increasing grass production with FRG will increase fertilization needs. These needs are partially fulfilled by the increase in excreta from grazing animals. This methodology offers a consistent framework but needs refinement to take into account indirect effects, such as the impact of FRG on white clover or other grassland species.

### 4.2. System Redesign

Two types of crossbreeding have been tested in this study to produce beef: crossing a continental beef breed with an early-maturing breed [14] and crossing a dairy breed with a Belgian Blue sire [22].

The Salers-Angus crossbreeding has been tested in the INRAe experimental farm of Laqueuille [12]. Crossbreeding a Salers cow with an Angus bull produces grass-fed young fattened animals. However, more harvested forages were required per animal since young bulls were kept one additional winter on the farm. This explains that high net HEP efficiency is obtained only for reduced herd size (Figure 6), allowing to interconnect animal needs to grassland resource availability. In addition to the FRG, when grazing is possible, the quality of the fodder harvested and stored must always be excellent to obtain sufficiently heavy and well-conformed animals. It was a challenge for the past years characterized by severe drought limiting forage self-sufficiency in the experimental farm. Furthermore, as this type of animal is far from the standards in the beef industry, the marketing of animal products was not always favorable. In fact, there have been instances where finished animals have been sold for less than if they were sold in foreign markets. The implementation of such system redesign does not only lead to modifications during the production phase but to modifications on the whole value chain up to the consumers, with a need to overcome some lock-ins at different levels [39]. While crossbreeding with an Anglo-Saxon breed was considered in this study, the potential of using early-maturing phenotypes within local breeds could be a lever to promote the adoption and the valorization of grass-based beef production.

The production of beef from the dairy herd offers the possibility to strongly reduce GHG emissions thanks to the low CO_2_ costs of the crossbred calves coming from the dairy herd, as keeping the mother cows is the dominant contribution to all impact in the suckler-based system [40]. Indeed, while having similar or even higher daily methane emissions than suckler cows, dairy cows produce high quantities of milk, which dilutes the GWP of the male calves produced. Pasture-based dairy-beef systems have also been studied in [16], where similar GHG emissions from 11 to 17.2 kg CO_2_eq/kg carcass are computed for the progeny of late maturing sires, depending on the slaughter age and the soil quality. The important change in the BE-SR scenario and the choice to keep a similar stable size, and therefore a limited stocking rate after the redesign, mainly explain the suitable net HEP efficiency. Indeed this stocking rate allows the production of grass and by-product-based meat with a low dependency to feed resources that could be valorized by humans. In [16], high net HEP efficiencies are observed only for older steers slaughtered at 28 months. Nevertheless, these steers have the highest GWP. In the BE-SR scenario, the grass and by-product-based bull system tested, with bulls slaughtered when 19 months, allows for combining lower GHG emissions and lower feed-food competition. Now, the possibility of achieving a sufficient fat score under such a production scheme with bulls requires experimental validation. Indeed, in [41], in Ireland, while crossbred steers with Belgian Blue obtained heavier carcasses with better conformation than Friesian or Aberdeen Angus crossbred, their carcasses were not acceptably finished off pasture at 19–20 months with rotational grazing. The possibility of using early-maturing sires could facilitate the fattening, with the drawback of a higher GHG emission per kg of carcass [16]. Adding a three-month period of fattening in the barn after the pasture, as studied in [11], showed acceptably finished carcasses, heavier than considered in this study. The German scenario, in which a reduced net HEP efficiency is observed, indicates potential drawbacks of the adoption of this new technology on the dairy farm without reducing the stocking rate in order to maintain the level of fodder autonomy.

The stocking rate range allowing for high net HEP efficiency is different in the BE (<2.8 LU/ha PG) and FR-IT (<1.1 LU/ha PG) system redesign (Figure 6). This difference could be explained by the availability of by-products, and the higher grasslands yield observed in Wallonia compared to the Massif Central [42]. This threshold is not reached in the German case, where 88% of the agricultural area is composed of tillable land leading to a high stocking rate per ha of grassland and, therefore, to the inclusion of a high share of human-edible protein in the beef diet.

While an important reduction of GHG emissions is observed per kg of carcass beef produced in the BE-SR scenario, the results do not transfer to farm-level emission since beef production per hectare has also doubled. As in the model, methane emissions are computed based on feed ingestion, which is similar between the baseline and the SR in the Belgian scenario. Such an innovation would only be beneficial to reduce GHG at the territorial level if it is associated with a reduction of the land used for beef production.

### 4.3. Relevance of the Modeling Framework to Redesign Production Systems

The modeling framework allows having a sustainability assessment related to innovation implementation. The number of indicators presented and discussed here has been limited to net HEP efficiency for feed-food competition and GWP for the environmental pillar of sustainability. The scope of the sustainability evaluation could also be enlarged while taking into account impacts such as acidification or eutrophication, as was performed in [23]. Increasing the number of indicators, as in [6], for instance, could improve the understanding of the impact of innovation at different scales.

While the quality of the output depends on the quality and the details of the assumption made in the modeling, some uncertainties remain as the ability to produce a carcass with a decent fat score in the BE-SR scenario. Those uncertainties are meant to be further explored during focus group discussions with farmers and other stakeholders, during which the current results are used as a base for discussion.

In this study, the impact on the Belgian or on the Italian fattening farms is not explored. Similarly, a generalization of the fattening of dairy crossbred bulls will have an impact on veal production in the EU. Widening the scope of the study is, therefore, necessary to provide a complete picture of the impact of such a redesign.

## 5. Conclusions

Using the farm-model FarmDyn innovations were tested on their contribution to the net HEP efficiency, GHG emissions, the work time necessary for production, and the farm profit. The introduction of FRG increased farm profit and work time. FRG associated with crossbreeding with early-maturing breeds and subsequent fattening on grasslands increases net HEP efficiency compared to intensive fattening of weanlings with maize silage, as observed in a French–Italian system. However, the effect is limited when connected to high stocking densities needing the importation of huge amounts of external feeds. Beef production based on dairy herds allows for significantly reduced GHG emissions compared to traditional suckler cow systems in the Belgian system tested. Nevertheless, a net protein efficiency higher than one also requires a stocking rate adapted to grassland production and the availability of by-products.

The results show potential pathways to improve the sustainability of current beef production systems at the farm level. Future research should focus on experiments with larger farm samples to validate the results given the high degree of variability found in beef production systems within each country and ultimately verify results in field trials on current farms.

## Figures and Tables

**Figure 1 animals-13-01020-f001:**
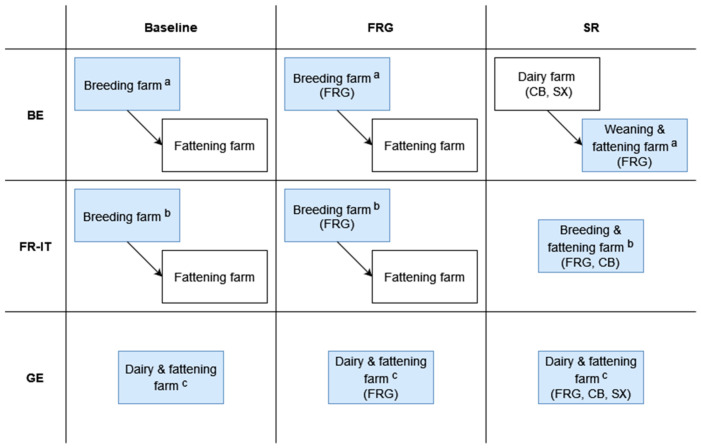
Overview of the beef production system considered in the study and the associated scenarios (baseline, fast rotational grazing (FRG), and system redesign (SR)). In the SR scenario, crossbreeding (CB) and sexing (SX) are applied in dairy farms. The farms of interest, labeled a in Belgium, b in France, and c in Germany (in blue), is the farm in which the tested innovation take place. Farm-level indicators, such as the profit, are computed for this particular farm.

**Figure 2 animals-13-01020-f002:**
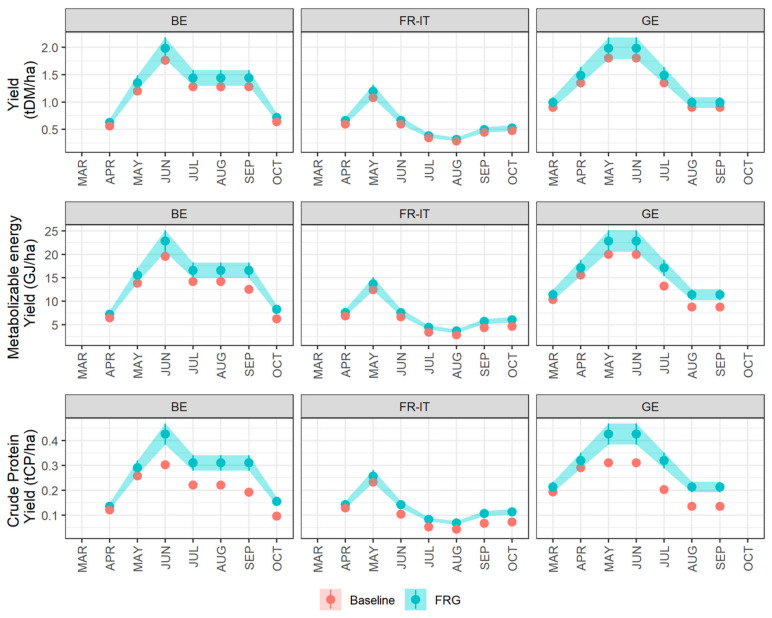
Dry matter, metabolizable energy and crude protein yield profiles for continuously (CG) and rotationally grazed (FRG) pastures in the three production systems.

**Figure 3 animals-13-01020-f003:**
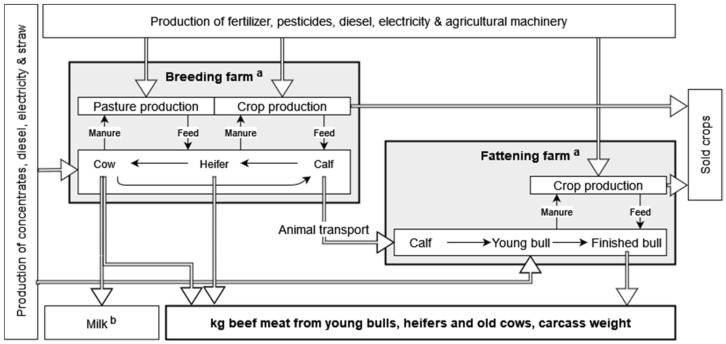
System boundaries of the analyzed beef production systems. Source: adapted from [23]. ^a^ in the German system breeding and fattening are integrated in one farm sparing animal transport. ^b^ milk is considered a co-product on the dairy farm of the German system and in the system redesign scenario in Belgium.

**Figure 4 animals-13-01020-f004:**
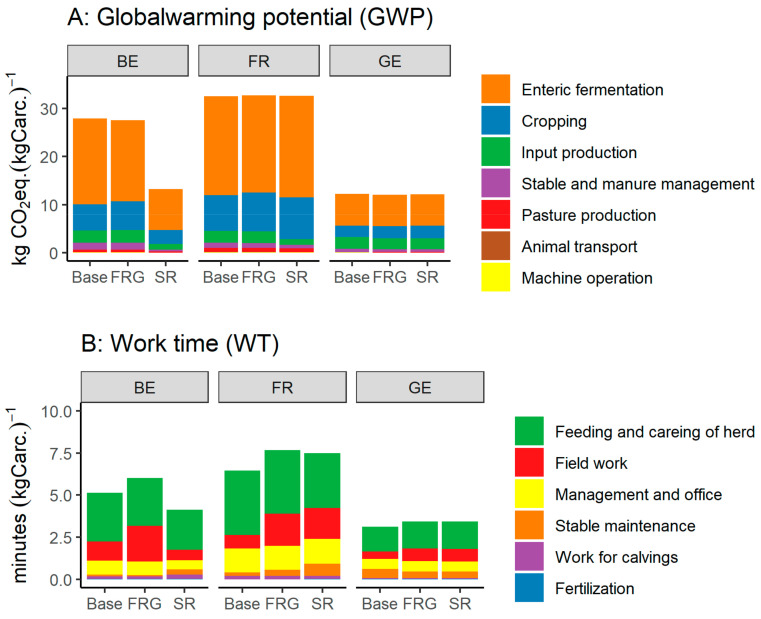
(**A**) GWP per kg of carcass and its different sources for the baseline (Base) and in the fast rotational grazing (FRG) and system redesign (SR) scenarios applied to the Belgian (BE), French–Italian (FR-IT), and German (GE) case studies. (**B**) Work time per kg of carcass and its different components.

**Figure 5 animals-13-01020-f005:**
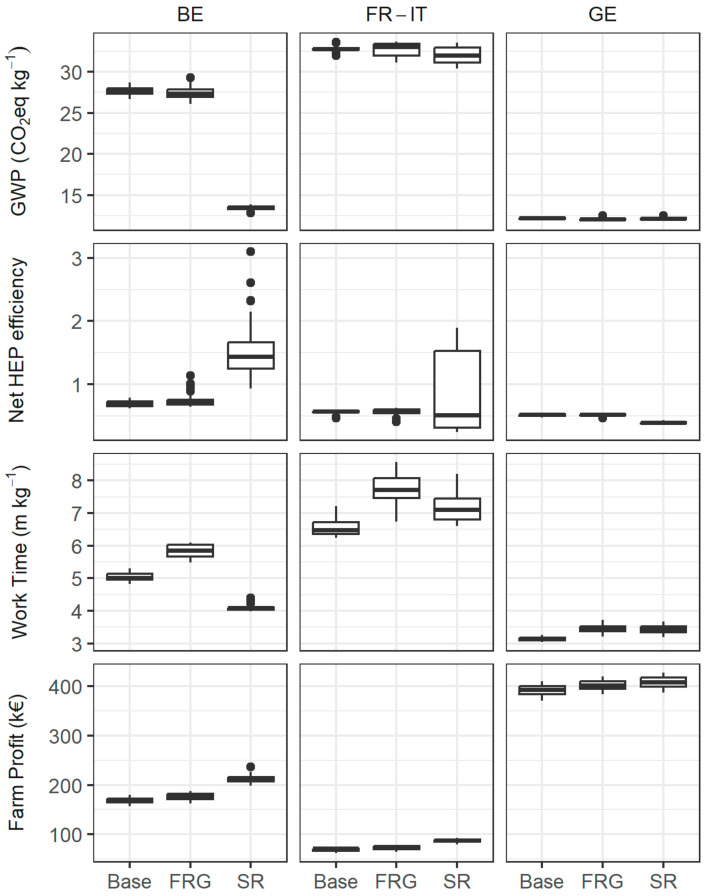
Results of sustainability evaluation of the scenarios supported by the sensitivity analysis for the baseline (Base) and the fast rotational grazing (FRG) and system redesign (SR) scenarios applied to the Belgian (BE), French–Italian (FR-IT), and German (GE) case studies. The variables studied are from top to bottom: the global warming potential for the production of 1 kg of beef carcass, the net human-edible protein efficiency, the work time to produce 1 kg of beef carcass, and the farm profit.

**Figure 6 animals-13-01020-f006:**
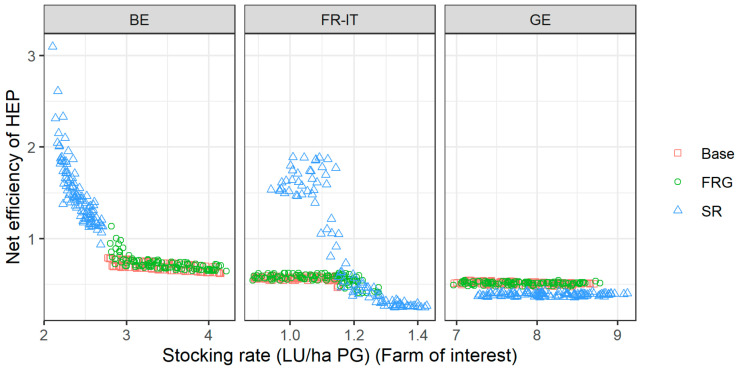
Relation between the net efficiency of HEP production and the stocking rate in livestock unit (LU) per ha of permanent grassland (PG) in the farm of interest for the tested scenarios for the baseline (Base) and in the fast rotational grazing (FRG) and system redesign (SR) scenarios applied to the Belgian (BE), French–Italian (FR-IT), and German (GE) case studies.

**Table 1 animals-13-01020-t001:** Overview of the baseline beef production systems [6].

System	BE		FR-IT		GE
Farm ^a^	BE-B	BE-F	FR-IT-B	FR-IT-F	GE
Country	Belgium		France	Italy	Germany
Location	Wallonia		Massif Central	Veneto	North-Rhine-Westphalia
No. males sold per year ^b^	78	120	38	227	56
No. of cows	155	-	79	-	130
Beef output (estimated carcass weight)	40,379	57,960	16,517	64,864	36,113
Breed	Belgian Blue		Charolais and Salers		Holstein
Arable land	54 ha	-	-	33 ha	198 ha
Grassland	64 ha	-	96 ha	-	27 ha
Other activities	Cash-crop production	-	-	-	Dairy and cash-crop productions

^a^ ”B” and “F” stand for breeder and fattener. ^b^ for breeding farms, this is the number of male calves sold, for fattening farms this is the number of slaughtered bulls.

## Data Availability

The data presented in this study are available on request from the corresponding author.

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
