# Peer review of "Exploring Rotational Grazing and Crossbreeding as Options for Beef Production to Reduce GHG Emissions and Feed-Food Competition through Farm-Level Bio-Economic Modeling"

_animals, 2023, doi:10.3390/ani13061020_

Round 1

Reviewer 1 Report

Manuscript ID: animals-2149503

Title: Exploring grass based beef production options to reduce GHG emissions and Feed-Food competition in EU through farm level bio-economic modelling

Authors: Alexandre Mertens, Lennart Kokemohr, Emilie Braun, Louise Legein, Claire Mosnier, Giacomo Pirlo, Patrick Veysset, Sylvain Hennart, Michaël Mathot, Didier Stilmant

Overall

This paper aimed to evaluate the impact of two “innovations” (rotational grazing of pasture & crossbreeding cattle) on “the emission of greenhouse gases (GHG), the net human edible protein (HEP) efficiency, the worktime used for production (WT) and the farms profit” of three farm systems, two of which included phases of the beef systems occurring in different countries.

I have a number of major concerns about the manuscript.

·         Overall, the manuscript is rather vague and difficult to follow - it is very challenging for the reader to understand exactly what is been compared and why.

·         The ‘novelty’ of the manuscript is not clear regarding what it contributes to the published literature.   

·         The selection criteria for the 3 ‘farms/ beef systems / (4) countries’ seems rather arbitrary.

·         It seems that two of the systems are based on farm data from Belgium, France and Italy as per the “SustainBeef project” (L105-107), yet there is no direct reference to the data ‘source’ for the third farm type in Germany (Table 1)? 

·         There are ‘confounding’ factors across the “farm systems” (countries) e.g. “manure” is solid in Belgium, France-Italy and liquid in Germany, which can have implications for interpreting the results obtained?

·         How are the “emissions” etc. partitioned across the farms/ countries…… etc.? This seems to limit the usefulness of the results?

·         The description of the Farm systems (‘control’ vs. “innovations”) needs to be explained more clearly.

·         Similarly, the selection criteria for the 2 “innovations” seems rather subjective.

·         The assumptions used for the “Fast Rotational Grazing” seem to be rather simplistic e.g. In relation to “grass quality improvement”, I don’t think that rotational grazing simply provides “spring quality all year” (Table 2) – the nutritive value of autumn grass is generally inferior to spring grass? There is published literature comparing rotational grazing vs. set-stocking…..etc. which would provide a more comprehensive biological basis for this “innovation”. It is not clear what the ‘control’ grazing management regime is? It is not clear to what ‘components/ phase/ segments’ of the systems the “Fast Rotational Grazing” was applied too?

·         There is insufficient information presented on the effect of “crossbreeding” on physical output. What was assumed and what was the basis for the assumption? Exactly how did the impact of crossbreeding (vs. pure bred?) manifest itself – heterosis vs. carcass composition…..etc. - and what were the magnitude of the effects? Again, there is copious published literature comparing purebred vs. crossbreeding …..etc. to provide a biological basis for this “innovation”.

·         There is insufficient information presented on the ‘inputs’ (physical, financial…..etc.) assumed in the model, and how and to what degree the systems ‘changed’ to accommodate the integration of the “innovations”.

·         Likewise, there is insufficient information presented on the ‘outputs’ produced for the systems.

·         In general, the results need to be presented in a more complete and logical fashion – the critical intermediary steps and outputs are ‘missing’.

·         It is not clear if these “innovative” production systems are biologically feasible e.g. Were the cattle on these systems ‘finished’ – adequate subcutaneous fat / carcass fat score? This is related to an earlier point – what biological data was the model based on in order to determine the impact of “innovation(s)”?

·         There is already a lot of published international literature evaluating the impact of ‘grazing’ in bull-beef systems……., comparing early-maturing vs. late-maturing sire breeds in grass-based beef systems………. etc. – this needs to be integrated, cited and discussed.

Reviewer 2 Report

Overall:

-          English needs revising in the whole document

-          The presentation of statistical analyses is poor

-          Description of methodology is lacking specifics

-          Discussion is incomplete

-          My expertise is not sufficient to judge about cross-breeding, beef fattening or profits

-          Overall the message is valid: using dairy calves instead of suckler calves reduces GHG emissions because dairy cows are productive. However, the methodology is inadequate.

Simple summary:

-          The statements given should be generally put into context of where the reduction is measured from. What is the baseline?

-          Sentences are over simplified so that the reader cannot get the overall message

-          Is ‘work time’ not included in profit calculations?

Abstract:

-          Again, 52 % reduction from which baseline? In comparison to what?

Introduction

-           

Materials and methods:

-          What is a systemic analysis?

-          How many farms are there from each farming system category? Is is just said ‘several’.

-          The fast rotational grazing scenario is questionable.

o   What does it mean exactly ‘fast rotational grazing? Is that possible to implement everywhere?

o   What is the comparison ‘continous grazong’? what does that mean exactly?

o   An estimate of 10 % increase in DM yield (??) based on an interview of a single person does not seem very scientific. That is also relevant for all other estimates set in the scenario.

o   Grass quality improvement: spring quality all year is very unspecific. Even in a ‘fast rotational grazing system with high nutritive value of the herbage there is a change in quality over the year. So just to assume spring quality all year, without classifying what that means exactly is not enough in my opinion.

o   The level of supplementary feed included in the animals diet (here it is said to be possible up to 30 %) highly influences the utilisation of the pastures in rotational grazing

-          Methodology for enteric methane: how can the grass quality have an effect on enteric methane if the methodology is based on DM intake (IPCC 2019)? Or is the equation for GE intake used?

-          What are shadow prices?

-          I wouldn’t mention farms profit and worktime, which are not related to environmental impact categories in the same sentence as GHG and HEP

-          Which type of by-products are used?

-          What is an ‘integrated dairy farm’? Do you mean to say ‘Integrated crop livestock system?

Results:

-          In the methods it was not mentioned that FRG requires more fertilizer. What was the baseline level of fertilizer? How much more was applied? When?

-          Can the effect of the single management practice FRG not be separated from the effect of higher fertilizer requirement? Just changing the grazing management will have an effect on sward quality regardless of fertilizer input level. It is unfortunate that the effect of grazing management cannot be assessed on its own in the way the study is designed.

-          The SR scenario is quite complicated and to understand the results the reader has to constantly go back to the method description

-          ‘anti-correlated’ should be reworded

-          Figure 2: how is the drastic change in BE from base to SR explainable when there is almost nothing happening in the other farms? How can especially the enteric fermentation be reduced by what looks like almost 50 % by a change in the system design?

-          Figure 2 needs explanations of the abbreviations FRG and SR, BE, FR and GE in the figure caption

-          p < 10-5 is not an acceptable presentation of a p value

-          I don’t understand the methodology of using a sensitivity analysis and the presenting single numbered results for each strategy without presenting variations and statistics

-          It makes no sense the implement FRG just for the dairy cows in GE. Why was it not applied for the beef cows as that is the focus of the study? It was also not said in the methods that that was the case. And the additional yield is used to save bought in concentrates: but just for the dairy cows right?

-          Figure 3: what means robustness of sustainability evaluation? Abbreviations same as for fig 2

Discussion

-          The increase of work time with FRG should be questioned. Other tasks get less instead (indoor feeding, cleaning cubicles etc.)

-          My critics on grass quality estimates remain, too unspecific

-          Discussion for FRG is very short. It should be included why the fertilizer is needed and if that is a reasonable assumption

-          Some discussion about FRG is located in the cross breeding paragraph. Structure of discussion should be revised

-          Just looking at GHG emissions, HEP and profitability does not justify calling the study a complete sustainability assessment. What about acidification, eutrophication, land use, nitrogen losses/footprint etc. Just don’t call it ‘complete’.

-          How can changes in beef production affect the ‘sustainability’ of crop production?

Conclusion:

-          The third sentence repeats the first sentence.

-          The conclusion that ‘The introduction of FRG increased farm profit and work time and decreased GWP’ is not based on the results. It was said that the effect of FRG on GWP was marginal (<2%).

Round 2

Reviewer 1 Report

Manuscript ID: animals-2149503 (REVISED)

Title: Exploring grass based beef production options to reduce GHG emissions and Feed-Food competition in the EU through farm level bio-economic modelling

Authors: Alexandre Mertens, Lennart Kokemohr, Emilie Braun, Louise Legein, Claire Mosnier, Giacomo Pirlo, Patrick Veysset, Sylvain Hennart, Michaël Mathot, Didier Stilmant

 Overall

The revised manuscript is clearer.

 Title

The title needs to be revised to more clearly reflect the focus of the manuscript - i.e. specifically mentioning the ‘factors’, “rotational grazing” & “crossbreeding”.

Introduction

Line 60-62: This manuscript is about beef production, not dairy cows?

L62-63: What does “modern breeds” and “less suited” mean? Please define and clarify, and provide supporting references.

L 64: Additional summary information is required on the “SustainBeef” project.  Specifically what did this project address? How many/which countries were involved? Ultimately need to know was the design of the “SustainBeef” project sufficiently robust to identify the 3 “typical farms” (production systems) and 2 “innovations” focused on in the current manuscript?     

References

It seems that many references are missing? In the reference list the number of references is ‘36’ BUT the citation number reaches ‘43’. Furthermore, there seems to be a mix-up between the reference list and the reference numbers cited in the document - many of the citations do not apply to topic e.g. Line 59-60, reference ‘9’, a beef production study, does not address “…..the amount of grazing in the EU is declining”;  Ref ‘15’, cited in Line 53 does not address “crossbreeding” or “heterosis”; Reference ‘24’, cited in Line 101 and 210, is a meat quality study, not a Farm systems / FarmDyn-related modelling study?

This needs to be rectified before the reviewing process can proceed further

Reviewer 2 Report

The paper has improved substantially.

Just some minor remarks remain:

I would still question the validity of the difference in Yield/quality between FRG and CG but I accept that this is as close as you could get within your methodological constraints.

I would not call a rotational grazing system with a residency time of three to five days a ‘fast’ rotation. In dairy systems residency time within rotational grazing systems is often 24 hours or even 12 hours. This is also highlighted in the chosen reference. I would suggest to just call it ‘rotational grazing’ and leave out the ‘fast’. May be confusing otherwise.
